# Influence of Carbon Nanoparticle Addition (and Impurities) on Selective Laser Melting of Pure Copper

**DOI:** 10.3390/ma12152469

**Published:** 2019-08-02

**Authors:** Suraj Dinkar Jadhav, Sasan Dadbakhsh, Jozef Vleugels, Johan Hofkens, Peter Van Puyvelde, Shoufeng Yang, Jean-Pierre Kruth, Jan Van Humbeeck, Kim Vanmeensel

**Affiliations:** 1Department of Materials Engineering, KU Leuven, Kasteelpark Arenberg 44, B-3001 Heverlee, Belgium; 2PMA, Department of Mechanical Engineering, KU Leuven & Member of Flanders Make, Celestijnenlaan 300, B-3001 Heverlee, Belgium; 3Department of Production Engineering, KTH Royal Institute of Technology, Brinellvägen 68, SE-10044 Stockholm, Sweden; 4Department of Chemical Engineering, KU Leuven, Celestijnenlaan 200 F, B-3001 Heverlee, Belgium; 5Department of Molecular Visualization and Photonics KU Leuven, Celestijnenlaan 200F, B-3001 Heverlee, Belgium

**Keywords:** selective laser melting, copper, carbon-mixed-copper, laser reflection, laser absorption

## Abstract

The addition of 0.1 wt % carbon nanoparticles significantly improved the optical absorption and flowability of gas-atomized copper powder. This facilitated selective laser melting (SLM) by reducing the required laser energy density to obtain 98% dense parts. Moreover, the carbon addition led to an in situ de-oxidation of the copper parts during the SLM process. The properties of the as-built copper parts were limited to a tensile strength of 125 MPa, a ductility of 3%, and an electrical conductivity of 22.7 × 10^6^ S/m, despite the advantageous effect of carbon on the powder characteristics and SLM behavior. The modest mechanical properties were associated with the segregation of carbon nanoparticles and other impurities, such as phosphorus and oxygen along grain boundaries of epitaxially grown grains. Whereas, the low electrical conductivity was mainly attributed to the phosphorus impurity in solid-solution with copper.

## 1. Introduction

Copper is a ductile metal with exceptional electrical (58 × 10^6^ S/m, which is equal to 100% International Annealed Copper Standard (IACS)) and thermal conductivity (400 W/(m·K)) [1]. Consequently, it is widely used in applications, where these properties are of prime importance, such as heat exchangers, heat sinks, etc. [2]. The heat transfer efficiency of such parts can be improved by redesigning these parts with fine features, rendering them a higher surface area for heat transfer [3]. However, conventional manufacturing techniques are unable to fabricate geometrically complex-shaped parts with fine features. Consequently, metal additive manufacturing processes, especially selective laser melting, are gaining interest due to their ability to fabricate geometrically complex-shaped parts with minimum waste of starting powder material [4]. Combining the ability to fabricate geometrically complex-shaped parts while using SLM and the very high thermal conductivity of copper, parts with greater heat transfer efficiencies could be fabricated [5].

Accordingly, Lykov et al. [6] and Trevisan et al. [7] attempted to fabricate copper parts by SLM while using a laser power of 200 W. However, a maximum relative density of 88% was obtained. Similar part densities were also obtained when Jadhav et al. [8] fabricated pure copper parts while using a laser power of 300 W. The lower part density is attributed to the inadequate fusion of copper powder particles. This incomplete fusion is caused by the high laser reflectivity of copper for infra-red radiation (particularly at 1060–1080 nm, which corresponded to the near-infra-red radiation of the most commonly used lasers in SLM) and the very high thermal conductivity that instantaneously dissipates the small absorbed heat [9]. To overcome the issue of insufficient melting and to obtain dense copper parts, few approaches have been proposed. These approaches are either related to the modification of the laser (power or wavelength) or the starting powder (change of alloy composition or surface modification).

Within the former approach, few researchers [8,10,11,12] employed a high-power 1 kW fiber laser and fabricated dense copper parts, exceeding 96% of the theoretical density (TD). However, damage to the optical mirror of the SLM machine was reported, which was induced by sustained copper back-reflections [8]. Hence, the use of high-power fiber laser to fabricate parts from highly reflective metals, such as copper and silver, is discouraged. Alternatively, Heussen and Meiners [13] and Trumpf [14] proposed using a green laser (515 nm), while Formalloy [15] incorporated a blue laser (450 nm) in their SLM machine to process pure copper. This change of the laser source (wavelength) is motivated by the higher laser absorption of copper at visible light wavelengths. The enhanced optical absorption for the visible electromagnetic radiation is due to the shift of the optical absorption mechanism from the intra-band to the inter-band electronic transition [16]. It is claimed that the production of a dense copper part can be realized by the use of green and blue lasers. However, mechanical and electrical properties of the copper parts that were manufactured while using these lasers are not yet disclosed.

In the latter approach, many researchers preferred to use pre-alloyed copper powders with slightly lower optical reflectivity when compared to pure copper. Accordingly, they could fabricate dense parts from Cu-Cr [17,18,19], Cu-Cr-Zr [20,21,22], Cu-Sn [23,24], and Cu-Zn [25] alloys. However, the thermal and electrical conductivity values of these parts are substantially lower when compared to pure copper parts in the as-built condition [8,26]. Moreover, an upcoming trend to fabricate copper parts by SLM with high thermal and electrical conductivity, is to coat or convert the outer surface of the virgin copper particles with/into a thin (<100 nm) surface layer with higher laser absorption [26], Therefore, the current paper proposes coating copper powder with carbon nanoparticles. Although the idea of coating the copper powder with carbon has been proposed before [27], no one yet reported on their SLM behavior or on the obtained mechanical and electrical properties. Additionally, chemical vapor deposition (CVD) was proposed as a coating method, which is costly and challenging to upscale.

Accordingly, the current investigation proposes an alternative approach of mixing carbon nanoparticles with copper powder particles in order to improve the powder optical absorption and its SLM behavior. At first, a methodology was developed to apply a uniform carbon nanoparticle coating on the surface of the copper powder particles while using mechanical mixing. Subsequently, the influence of carbon addition on the powder properties, their SLM behavior, and the resultant SLM part properties is evaluated. Finally, a correlation is established between the microstructure, powder chemical composition, and the part properties.

## 2. Materials and Methods

### 2.1. Starting Powders and Their Characteristics

Gas atomized copper powder (99.7% purity, D50: 43 µm, LPW Technologies Ltd., Widnes, UK) and carbon black nanoparticles (Printex^®^ U, D50: 25 nm, Dakota Coatings NV, Belgium) were used as the starting materials. The particle size distribution (D50 value) for pure copper was measured while using an LS 13 320 laser diffraction particle size analyzer (Beckman Coulter, Inc., Brea, CA, USA) in dry-mode, whereas the D50 value for the carbon nanoparticles was obtained from the supplier test certificate. The copper powder was mostly spherical with a limited number of adherent satellite particles, as shown in Figure 1a. The copper powder was dry mechanically mixed with 0.1 wt % carbon nanoparticles for 12 h in an argon-filled polyethylene bottle while using a multidirectional mixer (Turbula T2F, Artisan^®^ Technology Group). A uniform distribution of nanometer-sized-carbon particles was observed on the surface of the copper powder particle (Cu + C0.1), as illustrated in Figure 1b. The amount of carbon was fixed at 0.1 wt % in order to (i) obtain a sufficiently high optical absorption (>60%) at the fiber laser wavelength, and (ii) in situ de-oxidize the copper part by inducing a chemical reaction between carbon and oxygen within the melt pool, which resulted in gaseous CO/CO_2_ formation. The chemical composition of the virgin copper powder was measured while using inductively coupled plasma optical emission spectroscopy (ICP-OES, Varian 720 ES) and the oxygen content was measured while using the inert gas fusion technique of LECO. The carbon content in the powder and the as-built parts was measured by coulometry (Coulomat 702 SO/CS). Table 1 provides the overall chemical composition of the carbon-mixed-copper powder before SLM.

The optical absorption and flowability of both virgin copper powder and the uniformly mixed copper-carbon powder were measured. The optical (laser) absorption was measured at room temperature while using a Perkin Elmer Lambda 950 UV/VIS spectrophotometer (Waltham, MA, USA). The measurements were performed in the full reflectance mode and spectralon^®^ was used as reference material with 100% reflectivity. The optical absorption was calculated from the reflectivity measurements while using the formula ‘optical absorption (%) = 100 − optical reflection (%)’. This expression assumes zero transmission and it is a reliable assumption in the case of opaque materials, such as copper powder. The flowability was evaluated while using the dynamic angle of repose (AoR) technique [28]. In this technique, a transparent cylindrical drum with a diameter and length of 55 mm and 15 mm, respectively, is half-filled with the powder. Subsequently, the drum was rotated at three rotations per minute (RPM) and a video was recorded for about 2 min. Finally, the angle between the powder-air interface line and a horizontal line was measured over time. The maximum angle just before sliding or tumbling of the powder from the drum wall is recorded as the avalanche angle. The average angle was calculated by averaging the minimum and maximum avalanche angles. A good flowability is defined by an average angle that is below 40°.

### 2.2. Selective Laser Melting

Selective laser melting (SLM) was carried out on an in-house developed machine at the Department of Mechanical Engineering, KU Leuven. The detailed scheme of the optical setup within this machine is presented in [29,30]. The machine was further upgraded by incorporating a continuous single-mode 1 kW fiber laser that has a wavelength of 1080 nm and a beam diameter of 60 µm (1/e^2^ value measured using BeamWatch^®^ AM) [31]. It can maintain an oxygen level below 100 ppm by filling the built chamber with a protective argon atmosphere during the complete build. At first, the STL files for cube-shaped parts, with dimensions (10 × 10 × 10) mm^3^, were prepared while using the Magics software, Materialise, Belgium. All of the cube-shaped parts were fabricated on a copper base plate applying various SLM processing parameters. The density of these parts was measured while using the Archimedes method and the relative part density was calculated while considering 8.87 g/cm^3^ as the theoretical (true) density, which was measured for Cu + C0.1 powder using Multipycnometer™ (Quantachrome Instruments). During parameter optimization, the laser power (*P*) was varied from 650 W to 900 W, the laser scan speed (*v*) from 400 mm/s to 1000 mm/s, and the hatch spacing (*h*) from 0.12 mm to 0.2 mm, while the layer thickness (*t*) was maintained as constant at 0.03 mm. Each layer was scanned once while using a bi-directional scan strategy and a 90° scan rotation was employed between the subsequent layers. Moreover, all of the SLM experiments were carried out without preheating of the base plate. The volumetric energy density (*e*) was calculated as ‘*e = P/(v × h × t)’* and reported in J/mm^3^.

The optimized SLM settings, yielding the highest part density, were selected to produce rectangular bars (60 mm × 10 mm × 2 mm) for electrical conductivity measurements. All of the rectangular bars were horizontally built and further machined to obtain dog-bone-shaped tensile specimens with a gauge length of 16 mm, a width of 4 mm, and a thickness of 2 mm.

### 2.3. Microstructural Characterization

Light optical microscopes (LOM) (Zeiss—Axioskop 40 Pol/40 A Pol and Keyence—3D Digital Microscope, VHX-6000, Mechelen, Belgium) and scanning electron microscopes (Nanosem 450, FEI and XL30 FEG, Philips, Eindhoven, The Netherlands) were used for the microstructural and elemental analysis. The scanning electron microscopes are equipped with a field emission gun and energy dispersive spectroscopy (EDS) detector. The parts that are required for the microstructural analysis and hardness were mounted in resin and ground on silicon carbide papers with grit sizes from 320 to 4000. Following the grinding steps, the parts were polished using a 3 μm diamond suspension and etched for 20 s in a solution that was composed of 3.5 g FeCl_3_, 2.5 mL HCl, and 75 mL C_2_H_5_OH solution.

### 2.4. Electrical Conductivity and Mechanical Property Characterization

Electrical conductivity (κ) was measured while using a four-point measurement technique (Resistomat^®^ model 2301-V001, Burster), and the obtained results are reported in S/m. Tensile testing was carried out on an Instron 4505 machine at a strain rate of 0.5 mm/min. The tensile parts were loaded perpendicular to the building direction (BD) and the proof (yield) strength (Rp0.2) at 0.2% strain, tensile strength (σ), and ductility (ε) are reported. The stiffness (*E*) was measured while using the impulse excitation technique (IET, IMCE, Belgium) according to the ASTM C1259-08 norm. Vickers micro-hardness measurements were performed on the polished parts while using a load of 0.3 kg and dwell time of 15 s (FV-700, Future-Tech Corp., Tokyo, Japan).

## 3. Results

### 3.1. Influence of Carbon Nanoparticle Addition on the Optical Absorption and Flowability of Copper Powder

Figure 2a shows the optical (laser) absorption of both powders (pure Cu and Cu + C0.1) at different wavelengths, which ranged from 400 nm to 1400 nm. The addition of carbon nanoparticles to the virgin copper powder increased the optical absorption significantly over the full range of measured wavelengths. At the fiber laser wavelength (within 1060–1080 nm), the optical absorption is increased from 29% for pure copper to 67% for 0.1 wt % carbon-mixed-copper powder. Figure 2b presents the continuously measured angles over a period of time during the dynamic angle of the repose test. The carbon-mixed-copper powder shows a smaller variation in the maximum and minimum angles when compared to the virgin copper powder, which indicated an improved powder flowability. The small angle of repose value or smaller variation between the maximum and minimum angle demonstrates that the powder easily rolls inside the drum during rotation. As both of the powders show a dynamic angle of repose below 40°, they exhibit an excellent flowability and they are considered to be suitable for SLM.

### 3.2. SLM Parameter Optimization and Carbon and Oxygen Content Evolution

Figure 3 shows the relative density (%) of the SLM parts versus the applied laser volumetric energy density (J/mm^3^). The various applied laser power levels are indicated while using different markers. The higher part densities are consistently observed for the parts processed using a power of 725 W for a wide range of energy densities (laser scan speeds). Therefore, an optimized parameter (*P* of 725 W, *v* of 400 mm/s and *h* of 0.12 mm) combination, rendering the highest part density exceeding 98%, was selected for the fabrication of tensile and electrical conductivity bars. It is important to note that the parts processed while using a laser power of 900 W exhibit consistently lower relative density values, whereas the parts that were processed using a relatively lower laser power of 800 W and 725 W exhibit higher part densities.

Table 2 shows the amount of carbon and oxygen in the carbon-mixed-copper powder (measured before and after the SLM process) and the corresponding SLM part. The measurements show a nearly 38% and 72% reduction in the amount of carbon and oxygen, respectively, in the manufactured part as compared to the powder after being used in SLM.

Moreover, a reduction in the carbon and oxygen content is also observed within both powders (before and after SLM powders), as shown in Table 2. This indicates that carbon and oxygen are not only removed from the SLM part, but also from the powder during the SLM process.

### 3.3. Microstructural and EDS Analysis

The microstructural characterization, including EDS analysis, was carried out on the top surface (XY plane) as well as on the vertical cross-section plane (YZ plane), where Z is considered to be parallel to the building direction (BD). The interrelationship between the location of impurities and the microstructure development was investigated and reported in this section.

#### 3.3.1. Microstructural Analysis of As-Built (Unpolished) and Polished Top Surfaces

Figure 4a,b shows SEM images of the top surface in the as-built (unpolished) condition. The shape of the melt pool and laser scanning direction (SD) are indicated in Figure 4a while using white dashed curved lines and arrows, respectively. The top surface of the part shows stable melt tracks mirrored over the melt pool fusion line, demonstrating the bi-directional scanning strategy that was employed during SLM. An overlap of 30–40% is observed between two neighboring melt tracks without any evidence of surface porosity being caused by lack-of-fusion defects. Nevertheless, carbon segregation (due to carbon floating on top of the copper melt) and surface cracks (indicated by red arrows) are observed on the top surface. The carbon segregation is visible along the melt pool borders, whereas the surface cracks are visible crossing the melt pool borders.

Figure 4c,d presents the polished top surface of the part. The surface cracks that were observed on the as-built top surface (Figure 4b) are located all over the surface, as shown in Figure 4c. However, a closer view of these cracks in Figure 4d showed that the cracks are filled (and closed) by segregated material. Accordingly, the surface cracks are termed segregation defects. The samples were etched to observe the orientation relationship between the grains and the segregation phenomenon. Figure 4e,f shows that the defects are present along the grain boundaries and they are filled with segregated material.

#### 3.3.2. Microstructural Analysis of Cross-Section Plane Parallel to Building Direction

Figure 5 shows the vertical cross-section of a 98% dense part that was processed using the optimized laser scan settings. Figure 5a,b shows the defects oriented parallel to the building direction (BD), as indicated by dashed red arrows. The grains with epitaxial growth are orientated parallel to the BD and the segregated material defects are located along the grain boundaries of these grains, as seen in Figure 5c,d. It should be noticed that these ‘cracks’ (as observed on the unpolished top surface) are actually filled with segregated material. Figure 5e,f shows that these defects can be as long as 2 mm, and they are oriented along the BD. At higher magnification, as shown in Figure 5e, it can even be seen that the segregation lines are aligned with the sub-grain structures, exhibiting a cellular solidification morphology of the epitaxial grains. Moreover, these defects are observed at the boundary between different sub-grain (cell) orientations. To be clearer (inset of Figure 5e), the black arrows indicate the orientation of solidification cells within one grain, while the black dot illustrates that other solidification cells, located in a neighboring grain, are pointing out of the plane. The backscattered electron (BSE) image in Figure 5f shows the chemical difference between the copper matrix and the region that was enriched with segregated material. The defects are darker in color as compared to the copper matrix, which indicates that lighter elements segregate to the intergranular regions.

### 3.4. Mechanical and Electrical Properties

Table 3 summarizes the mechanical and electrical properties of as-built SLM parts, which were processed from carbon-mixed-copper powder, and compares them to the properties of oxygen-free copper in the soft condition for reference [32]. In general, lower mechanical and electrical properties are obtained for the carbon-based copper parts when compared to the conventionally processed pure copper in the soft condition. Based on the electrical conductivity (22.7 × 10^6^ S/m or IACS of 39.2%) measurements, the thermal conductivity of the part can theoretically be calculated while using the Wiedemann-Franz law [33]. When considering a Lorenz number (L) = 2.41 × 10^−8^ WΩ/K^−2^ [1] and temperature T = 300 K, the calculated thermal conductivity equals to 164 ± 2 W/(m·K). The thermal conductivity of the carbon-mixed-copper is comparable to the highest conductivity of SLM processable materials, such as AlSi10Mg [5].

#### EDS Analysis of the Segregation Defects

EDS analysis was performed in the segregated defect area and its composition was compared to the copper matrix. Figure 6 demonstrates that the segregated area contains a larger amount of impurity elements, such as carbon, phosphorus, and oxygen, when compared to the copper matrix and the copper is present in the balance amount. All of the defect features are dominated by carbon, followed by oxygen and phosphorus, as they have piled up in the segregated regions.

### 3.5. Fracture Analysis of the Tensile Bars

Fracture analysis of the tensile bars was performed to establish a correlation between the microstructure and mechanical properties, as shown in Figure 7 and Table 3. More specifically, Figure 7a,b shows the absence of plastic deformation (necking) near the fractured plane, which indicates a brittle fracture. On the fractured plane, a brittle morphology is clearly visualized along with intergranular cracking, as indicated using white arrows. Few surface cracks parallel to the building direction are observed near the fracture plane, as shown in Figure 7c. The direction of the tensile loading is indicated while using a white double-sided arrow and it is perpendicular to the orientation of the surface cracks. The width of the cracks that were observed near the fractured surface is larger and the crack width could have extended further during tensile testing. Figure 7d shows multiple defects near the fractured surface, as indicated by red arrows. The defects are oriented parallel to the BD and they are situated at the grain boundaries of the epitaxial grains, as denoted by black arrows. The defects are observed at higher magnifications in Figure 7e,f and are composed of light elements. EDS analysis of the dark phase reveals the presence of carbon, phosphorus, and oxygen, along with a balanced amount of copper.

## 4. Discussion

### 4.1. Influence of Carbon Nanoparticle Addition on Powder Characteristics and SLM Processability

The addition of carbon nanoparticles significantly improved the optical absorption of copper powder. This is due to the higher optical absorption of carbon for infra-red radiation. Moreover, a uniform distribution of carbon nanoparticles on the surface of the copper powder particles reduces the true contact area between the particles. This lowers the inter-particle cohesive forces between individual copper particles; hence, the powder flowability is improved [34,35].

Figure 3 showed that the dense SLM parts can be obtained from the 0.1 wt % carbon-mixed-copper starting powder, within a broad range of energy density values (200–500 J/mm^3^) while using a laser power of 725 W, whereas Jadhav et al. [8] showed that a minimum energy density of 700 J/mm^3^ is required to obtain dense SLM parts from pure copper powder when the same fiber laser was used. This successful fabrication of dense copper parts by SLM will enable the manufacturing of complex heat transfer devices [36], such as heat exchangers, heat sinks, heat pipes, magnetic inductor coils [37], and filters with lattice structures [20], with enhanced heat transfer efficiency.

A comparison of the SLM behavior of carbon-mixed-copper powder with pure copper powder [8] confirms that the mixing of carbon nanoparticles with copper is beneficial for reducing the required energy density to process copper by SLM. The reduction in the required energy density could be attributed to the fact that carbon nanoparticles float on the surface of the copper melt and that the carbon-coated-copper particles, originating from the powder bed neighboring the molten track, are continuously incorporated into the melt pool during the SLM process [38]. Moreover, carbon from the copper powder reacts with oxygen and evaporates as CO/CO_2_ gas. Consequently, a lower amount of carbon and oxygen is observed in the powder after SLM and the corresponding SLM part, as compared to the virgin Cu + C0.1 powder before SLM. It is important to note that the evaporation of elemental carbon is not expected, because of: (i) its high sublimation temperature (3630 °C, at atmospheric pressure) as compared to all other elements [39], (ii) the maximum temperature that can be achieved in the copper melt is lower than 2600 °C, even though a laser power of 800 W, scan speed of 300 mm/s and hatch spacing of 0.05 mm is applied [11], and(iii) the formation of carbon dioxide/monoxide (gas) is favored when the temperature of the system containing Cu, O, and C elements exceeds 150 °C, as per the Ellingham diagram.

The parameter optimization in Figure 3 showed that lower part densities were obtained at the highest applied laser power of 900 W, while a laser power of 725 W resulted in dense parts. This result can be explained by comparing the microstructures that were obtained for two parts processed at a constant laser scan speed and hatch spacing of 400 mm/s and 0.12 mm, respectively, yet with different laser powers of 900 W and 725 W. As seen from Figure 8a,b, the part processed using a laser power of 900 W shows extremely deep melt pools (~730 µm), along with keyhole pores. In contrast, the parts processed while using a laser power of 725 W exhibit a typical conduction-controlled semi-elliptical melt pool shape without any obvious porosity (segregation is still visible), as shown in Figure 8c,d. Therefore, the lower part densities that were obtained for the parts processed at a laser power of 900 W could be attributed to the keyhole phenomenon, trapping gaseous bubbles due to the rotational flow pattern forming a vortex at the bottom of the melt pool [40]. This phenomenon is commonly observed in the SLM process, when very high laser energy densities are employed, creating very deep melt pools [41]. Moreover, a lower part density could also be obtained when the applied laser energy density is not high enough to cause complete fusion of the powder particles (the part processed using 181 J/mm^3^ (*P* of 650W, *v* of 1000 mm/s and *h* of 0.12 mm) showed a relative density of 94.7%). In this case, the lower part density could be attributed to the presence of the so-called lack-of-fusion defects [42].

### 4.2. Segregation Defects and Their Effect on Mechanical and Electrical Properties

The defects that were observed in the parts are mainly caused by the segregation of impurities, such as carbon, phosphorus, and oxygen. Carbon was intentionally added for improving the optical absorption of the copper powder, whereas phosphorus and oxygen were already present in the copper starting powder.

The intentionally added carbon nanoparticles are consumed through two mechanisms. Firstly, they can react with oxygen to evaporate as CO or CO_2_ by an in situ reaction [43], which is feasible during a typical SLM process. Secondly, the carbon nanoparticles are pushed ahead of the solid-liquid interface into the liquid melt during solidification due to their poor solid solubility in solid copper (up to 7 ppm) [44] and poor wettability [45] with liquid copper. Due to these characteristics, the addition of carbon should not harm the electrical conductivity of copper. However, the carbon nanoparticles can segregate along the melt pool boundaries, including the top surface of the melt pool, as well as at the grain boundaries inside the part due to carbon solute rejection and poor carbon wettability [45]. Both segregation phenomena result in the formation of carbon-rich features with a crack-like appearance when they are observed from the top surface of the sample, as shown in Figure 4a,b. Within the cross-sectional view, one can see that these crack-like features consist of segregated impurity elements, indicating that the defects observed in the part are mainly due to the segregation of impurities, such as carbon, phosphorus, and oxygen.

Conventionally, phosphorus is added to copper for de-oxidation and strengthening (especially creep) purposes. When copper, alloyed with phosphorus, is heated above its melting point during casting, the intentionally added phosphorus reacts with oxygen, forming phosphorus oxide. The newly formed phosphorus oxide has a very low density of 2.39 g/cm^3^. Consequently, it segregates/floats on top of the liquid in the form of slag that can be skimmed off as dross. This results in a reduced impurity content (phosphorus and oxygen) in copper after casting, and a high electrical conductivity is obtained in the as-cast part. However, there is no possibility for removing such phases during the SLM process; hence, they are incorporated inside the SLM part, forming defects. Consequently, the carbon-mixed-copper part retained 0.17 wt % of phosphorus in its as-built state. Note that the presence of 0.17 wt % phosphorus can reduce the electrical conductivity of copper by 60% [46], due to its ability to form solid-solution with copper up to 0.5 wt % at room temperature [47]. Besides its detrimental effect on the electrical conductivity, phosphorus can also form very corrosive and toxic solid phosphorus oxides, such as P_4_O_10_ and P_4_O_6_, respectively. Additionally, gaseous phosphorus pentoxide (P_4_O_10_) is also toxic. Even though no gases were observed or detected during the SLM process, a safety precaution is recommended when working with phosphorus-containing copper alloys.

Oxygen, on the other hand, has an extremely low solid-solubility (up to 2 ppm) in copper [47]. Therefore, its potential negative influence on the electrical conductivity of copper is negligible, when compared to phosphorus [26].

Based on the above explanation, it could be stated that the poor electrical conductivity of 22.7 × 10^6^ S/m (or 39% IACS) is mainly caused by the formation of a copper-phosphorus solid-solution after SLM. Other factors, such as grain size, residual stresses, residual porosity, carbon, oxygen, copper-tin solid-solution impurity, etc., may also contribute to the lowering of electrical conductivity, but they are certainly not as influential as the copper-phosphorus solid-solution impurity [48]. Besides the aforementioned material issues, the tensile loading direction perpendicular to the orientation of the defects could also contribute to the premature failures by concentrating the stresses along the defects.

## 5. Conclusions

The SLM of carbon-mixed-copper powder was evaluated in this research. Mixing of carbon nanoparticles with gas-atomized copper powder improved the powder properties, such as optical absorption and powder flowability. The modified copper powder with the enhanced optical absorption facilitated SLM processing, resulting in the formation of nearly dense (98% TD) copper parts at relatively lower laser energy densities (200–500 J/mm^3^). Moreover, the carbon addition promoted in situ oxygen removal and purified the SLM part as compared to the virgin powder feedstock. However, the fabricated parts exhibited modest tensile and electrical properties. The microstructural analysis demonstrated a clear interrelationship between the initial chemical composition of the powder and the presence and orientation of a segregated phase, which was rich in phosphorus, oxygen, and carbon, on one hand, and the resultant electrical and mechanical properties, on the other hand. The low electrical conductivity was mainly caused by the presence of a phosphorus impurity, forming a solid-solution with copper. The lower tensile properties were linked to the segregation of carbon, oxygen, and phosphorus-based impurities, as well as their orientation along the grain boundaries of the grains with epitaxial growth.

Consequently, this research confirms the significant influence of impurity (alloying) elements present in the commercially available copper starting powders, on the properties of SLM parts. Moreover, the possibility of phosphorus oxide (pentoxide or trioxide) formation during the SLM process highlights the safety concern that is related to the use of phosphorus-containing copper powders. Accordingly, this research recommends the use of phosphorus-free copper powder during the additive manufacturing processes.

Furthermore, the in situ de-oxidation feature and the improved optical absorption offered by the addition of carbon nanoparticles could be seen as a cost-effective solution for minimizing the oxygen content in the copper part and as a method for reducing the risk of damaging the optical mirrors of the SLM machine. Finally, the issue of carbon segregation could perhaps be minimized by the addition of carbon-absorbing elements, such as chromium, titanium to copper. As such, copper parts with a lower amount of impurity elements and improved heat transfer efficiencies could be fabricated.

## Figures and Tables

**Figure 1 materials-12-02469-f001:**
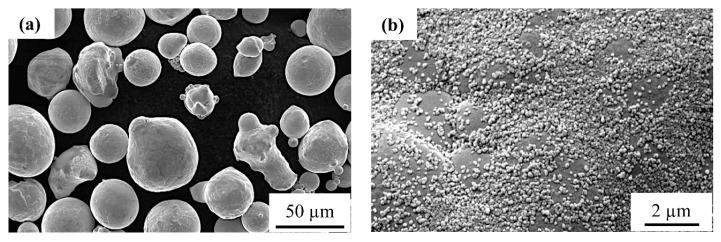
(**a**) Secondary electron image of as-received copper powder (virgin) before the addition of the carbon nanoparticles, (**b**) Uniform distribution of carbon nanoparticles on the surface of a copper powder particle.

**Figure 2 materials-12-02469-f002:**
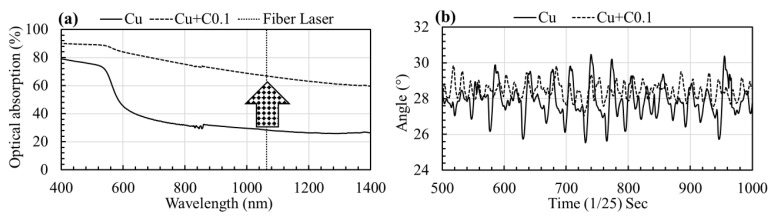
(**a**) Optical absorption (%) for both pure copper and 0.1 wt % carbon-mixed-copper powder within the 400–1400 nm wavelength range. A significant increase in the optical absorption, after addition of carbon nanoparticles to the copper powder, is evident at room temperature, (**b**) the carbon-mixed-copper powder shows less variation in the maximum (avalanche) and minimum angle, indicating an improved powder flowability.

**Figure 3 materials-12-02469-f003:**
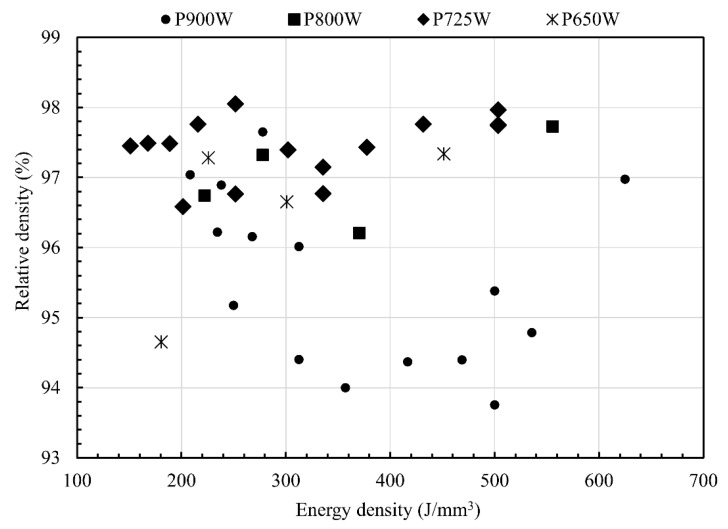
The relative density of SLM parts versus the applied laser volumetric energy density showing higher relative part densities could be obtained for a wide range of energy densities when a laser power of 725 W is employed.

**Figure 4 materials-12-02469-f004:**
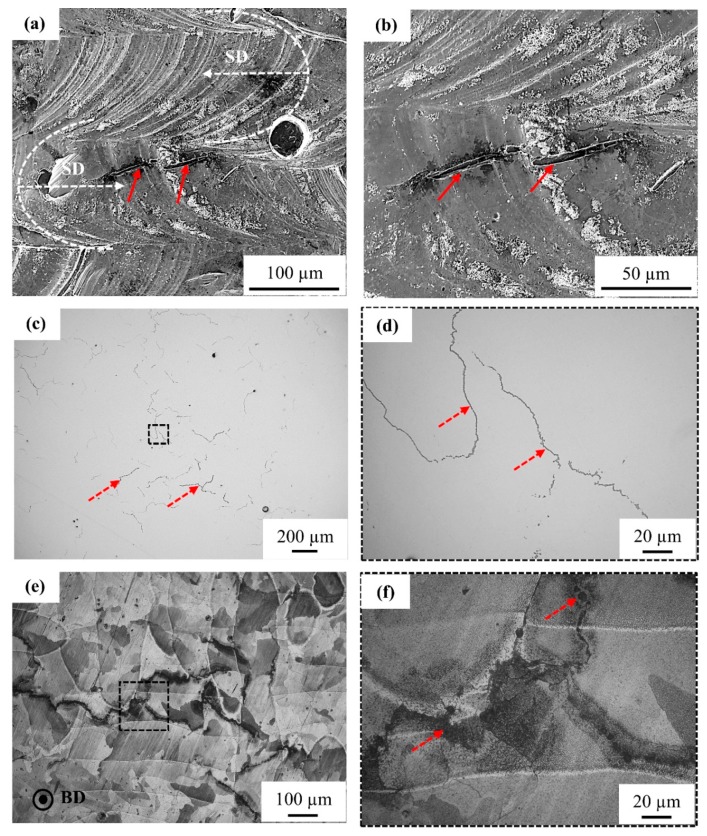
(**a**,**b**) SEM images of the top surface of the dense part processed with the optimized SLM settings. Melt pool boundaries and laser scan (SD) direction are indicated using white dashed curved lines and arrows, respectively. Surface cracks observed on the part top surface are indicated using red arrows, (**c**,**d**) LOM images of the XY plane after polishing. The surface cracks observed on the top surface are filled with a segregated material, as indicated by red dashed arrows. The defects seem to be located all over the surface forming a closed-loop in some areas, (**e**,**f**) LOM image of the polished sample in etched condition. The defects are observed along the grain boundaries growing over multiple melt pools.

**Figure 5 materials-12-02469-f005:**
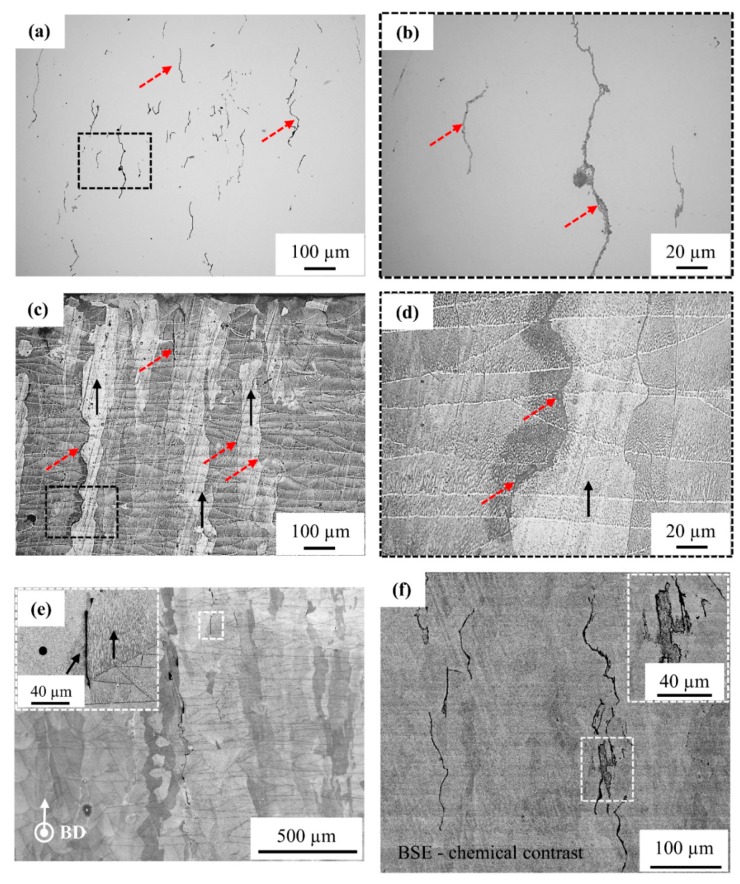
(**a**,**b**) LOM image of the vertical cross-section of the part in the un-etched condition. The part was processed using the optimized laser scan setting yielding a dense part with defects caused by segregation. All the defects are orientated parallel to the building direction, (**c**,**d**) OLM images of the vertical cross-section of the part in etched condition, the red dashed arrows indicate the defects caused by segregation, while solid black arrows indicate the grains with epitaxial growth. The defects are observed at the grain boundaries of the grains with epitaxial growth, (**e**) SEM image showing the location of the defects, the inset shows that the defect is present along the grain boundary of the grains with epitaxial growth. The cells are oriented parallel to building direction (BD) inside the epitaxial grains. The black arrow and black dot indicate the direction of cells, (**f**) Backscattered electron- scanning electron microscopy (BSE-SEM) image of the part showing chemical contrast, the dark contrast of the defect indicates the presence of light elements.

**Figure 6 materials-12-02469-f006:**
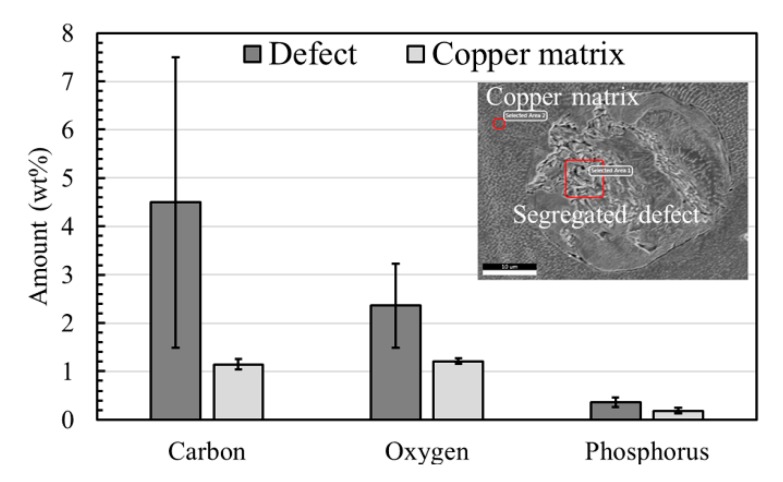
Energy dispersive spectroscopy (EDS) analysis comparing the chemical composition of the defect area and the copper matrix. The defect area shows a higher amount of impurities compared to the parent copper matrix. In all cases, copper is present as a balance element.

**Figure 7 materials-12-02469-f007:**
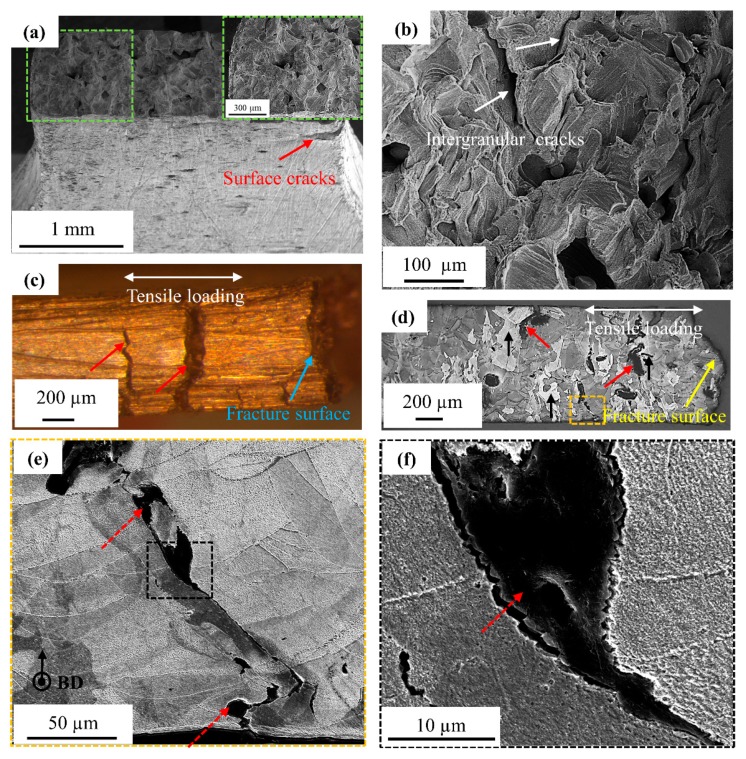
(**a**,**b**) SEM images of a fractured surface showing brittle fracture along with intergranular cracking. A surface crack is also indicated by the red arrow, (**c**) Binocular image of the side surface of the tensile tested sample with surface cracks parallel to BD, (**d**–**f**) show LOM and SEM images of the cross-section of the fractured sample in etched condition. These images show that the defects are oriented parallel to BD and are situated at the grain boundaries of the grains with epitaxial growth. The sample was tensile loaded in a perpendicular direction to the defects. The defect-rich region contains light elements, such as carbon, oxygen, and phosphorus.

**Figure 8 materials-12-02469-f008:**
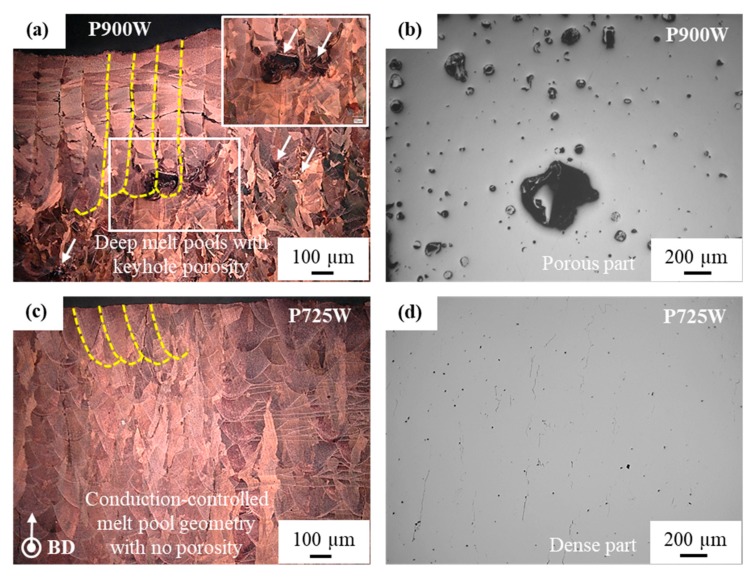
(**a**,**b**) Optical images of the part processed using a laser power of 900 W in etched and un-etched condition, respectively. The part shows extremely deep melt pools (~730 µm, indicated using yellow dashed curves) along with keyhole porosity lowering the part density. (**c**,**d**) show the microstructures of a part processed using a laser power of 725 W, indicating conduction-controlled semi-elliptical melt pools without obvious porosities. Both parts were processed at the same laser scan speed of 400 mm/s and hatch spacing of 0.12 mm.

**Table 1 materials-12-02469-t001:** Chemical composition of the virgin carbon-mixed-copper powder (before selective laser melting (SLM)).

Element	Amount (wt %)
Phosphorus	0.17
Tin	0.06
Oxygen	0.0925
Carbon	0.1
Copper	Balance

Note: 0.1 wt % of carbon was manually added during dry mechanical mixing.

**Table 2 materials-12-02469-t002:** Carbon and oxygen content in Cu + C0.1 powder before (virgin) and after SLM and within the corresponding as-built SLM part.

Element	Virgin Powder (Before SLM)	Powder (After SLM)	SLM Part
Carbon (wt %)	0.1	0.088	0.055
Oxygen (wt %)	0.0925	0.0515	0.0146

**Table 3 materials-12-02469-t003:** Mechanical and electrical properties of carbon-mixed-copper parts in the as-built condition and conventional properties of pure copper in the soft condition provided for reference [32].

Property/Alloy	P (wt %)	O (wt %)	Hardness (GPa)	*E* (GPa)	R_p0.2_ (MPa)	σ (MPa)	ε (%)	κ (S/m)
Cu + C0.1 (AB)	0.17	0.0146	0.637 ± 0.02	105 ± 2	64 ± 7	125 ± 11	3 ± 1	22.7 × 10^6^
Reference Cu-OF [32]	-	0.001	-	117	69	179–262	35	58 × 10^6^

Abbreviations: E: Young’s modulus, R_p0.2_: Proof(Yield) strength, σ: Tensile strength, ε: Ductility (Strain at break in %), κ: Electrical conductivity, AB: As-built, and OF: Oxygen (and phosphorus) free. All reference properties of Cu-OF are reported for the soft condition.

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
