# Peer review of "Influence of Carbon Nanoparticle Addition (and Impurities) on Selective Laser Melting of Pure Copper"

_materials, 2019, doi:10.3390/ma12152469_

Round 1
Reviewer 1 Report
In this manuscript, the authors studied the influence of carbon nanoparticle addition (and impurities) on selective laser melting of pure copper. Overall, the paper is interesting but require further modifications:
1- Can the authors show high-magnification images for fractography analysis?
2- Did the authors expect nanopartricles to got evaporated during SLM processing?
3- The segregation phenomena, including in-situ reaction, during SLM processing of nanoparticles reinforced metals was discussed by previous researchers. The authors need to go through the following example papers and cite them in order to provide link between their findings and current work:
- Thermal behavior and densification mechanism during selective laser melting of copper matrix composites: simulation and experiments
- Densification behavior, microstructural evolution, and mechanical properties of TiC/316L stainless steel nanocomposites fabricated by selective laser melting
- In-situ formation of novel TiC-particle-reinforced 316L stainless steel bulk-form composites by selective laser melting
Author Response
Please refer attachment

Reviewer 2 Report
The paper presents research in the actual and modern research field. The authors propose to use carbon nanoparticles to improve laser absorption properties of copper powder, which reflects a laser beam and can destroy laser optics (f-theta lens). Meanwhile, the results of the research cannot be called sufficient as due to a particular interaction between oxygen and phosphorus, which creates agglomerations of toxic phosphorus oxide (it is not written, but I suppose that they received trioxide P4O6), copper‐tin solid‐solution, copper‐phosphorus solid‐solution, other impurities. The authors recommend using copper powder without phosphorus.
The article has a few severe issues, which must be regulated before publishing.
1) The introduction must present a more detailed overview of the current state of the research. In the current form, it is not acceptable as it presents three not related to each other parts, which describes shortly copper, SLM, and carbon particles. The introduction should lead the reader from the known statement to the current problem of the industry (or actual scientific problem), there is recommended to formulate the actuality of the work, its scientific novelty, and scientific tasks to solve the current problem.
2) The section of Materials and Methods should be reorganized to make it better seen the main step of the study:
1) research the copper powder and carbon nanoparticles, the method of its mechanical mixing (a type of the equipment, if it was used);
2) features of working with SLM equipment (if it was a specially developed construction, it is better to show a scheme and a general view of it), the features of work preparation process (drying and preheating of the powder mix in the vacuum four, if it was done; preheating of the working platform; how was controlled atmosphere in the chamber), a short description of the obtained samples, which were produced and studied further;
3) characterization of the samples.
3) Did the authors a granulomorphometric analyze of powders? If not, why.
4) The obtained cross-section of the molten pool showed its elongated and disproportional shape. Did the authors analyze the possibility to use more proportional sizes for laser beam diameter (60 µm) and the thickness of the powder layer of 1000 µm? Did the authors consider a possibility to redistribute the energy in the laser beam as it was proved that equally redistributed energy in the laser beam spot gives better energy absorption by powder and more equally homogenized structure of the samples?
5) The toxicity and corrosion properties of the formed forms of phosphorus oxide must be discussed.
6) It is recommended to use internationally accepted symbols and units according to international standards SI for the electrical and mechanical properties mentioned in the article and particular in Table 3. For example, by Greek letter σ, https://en.wikipedia.org/wiki/Ultimate_tensile_strength, or electrical conductivity as it should be https://en.wikipedia.org/wiki/Electrical_resistivity_and_conductivity. Probably it is better to look carefully at the works of other authors or authors' guidelines for Materials journal.
7) The presentation of the numbers must be done according to international standards. Form of 2.41E‐08 is not acceptable.
8) The numbers of the paragraphs must be checked: 3.3, next 3.1.1.
9) The possibility to approximate data presented in Fig. 3a by a function must be discussed; the current presentation is not readable and informative.
10) The conclusions must include not only the main recommendations but also the practical significance of the research and further outlook. Just recommendation on avoiding copper powder with the content of phosphorus is not enough.
11) The list of references must be revised. The list of literature must present an international overview on the research subject and includes at least 30-50 positions (now there are a few works published in well-known international journals, some technical data sheets, advertising articles, which looks quite strange in the scientific article for publication in the authoritative pre-reviewed journal as Materials), 50% of them must be related to the works published in 2017-2019. It is recommended to mention at least 2 articles related to the works published in Materials.
12) English must be lightly checked as it has a few misprinting.
Author Response
Please refer the attachment

Round 2
Reviewer 1 Report
Well-done.
Author Response
Dear Reviewer,
We greatly appreciate your time and the quality feedback that you gave on our manuscript during the peer review process. Your feedback helped us to further strengthen the content of the manuscript.
Sincerely,
S. D. Jadhav, S. Dadbakhsh, J. Vleugels, J. Hofkens, P. Van Puyvelde, S. Yang, J-P Kruth, J. Van Humbeeck, K. Vanmeensel,
Reviewer 2 Report
The authors thoughtfully reworked the manuscript according to the comments, but still were not available to formulate for which use they need to process copper powder with the laser of 1kW. The questions of safety were highlighted, that is important. English must be checked. The manuscript can be accepted in the present form if the authors can give an example of parts, which can be produced from the copper powder with the laser power of 1 kW (actuality of the research).
Author Response
Please refer the attachment
